# Inhibitory Effect of Osthole from *Cnidium monnieri* on Tobacco Mosaic Virus (TMV) Infection in *Nicotiana glutinosa*

**DOI:** 10.3390/molecules25010065

**Published:** 2019-12-24

**Authors:** Ya-Han Chen, Dong-Sheng Guo, Mei-Huan Lu, Jian-Ying Yue, Yan Liu, Chun-Ming Shang, De-Rong An, Ming-Min Zhao

**Affiliations:** 1College of Horticulture and Plant Protection, Inner Mongolia Agricultural University, Hohhot 010019, China; yhchen1018@nwafu.edu.cn (Y.-H.C.); yuejianying2018@163.com (J.-Y.Y.); 2College of Plant Protection and State Key Laboratory of Crop Stress Biology for Arid Areas, Northwest A&F University, Yangling 712100, China; gds1995908@163.com (D.-S.G.); lu_meihuan@sina.com (M.-H.L.); 3Microbial Resources of Research Center, Microbiology Institute of Shaanxi, Xi’an 710043, China; 4Academy of Agriculture science in Baotou, Baotou 014010, China; liuyanww@126.com (Y.L.); chunmingsh@163.com (C.-M.S.)

**Keywords:** *Cnidium monnieri*, osthole, tobacco mosaic virus, half-leaf method, inhibitory

## Abstract

The coumarin compound of osthole was extracted from *Cnidium monnieri* and identified by LC-MS and ^1^H- and ^13^C-NMR. Osthole was tested for anti-virus activity against tobacco mosaic virus (TMV) using the half-leaf method. The results showed that stronger antiviral activity on TMV infection appeared in *Nicotiana glutinosa* than that of eugenol and ningnanmycin, with inhibitory, protective, and curative effects of 72.57%, 70.26%, and 61.97%, respectively. Through observation of the TMV particles, we found that osthole could directly affect the viral particles. Correspondingly, the level of coat protein detected by Western blot was significantly reduced when the concentrations of osthole increased in tested plants compared to that of the control. These results suggest that osthole has anti-TMV activity and may be used as a biological reagent to control the plant virus in the half-leaf method.

## 1. Introduction

Tobacco mosaic virus (TMV) belongs to the genus *Tobamovirus* and is transmitted by mechanical inoculation and insects with chewing mouthparts in a propagative manner [1,2]. TMV is an economically and destructively important plant virus with a wide host range, infecting more than 400 plant species from 36 families [3]. Recently, a survey of plant viruses was collected from 31 provinces in mainland China over a period from 2013 to 2017, which included over 41,000 vegetable crop samples from the *Solanaceae*, *Cucurbitaceae*, *Leguminosae*, and *Cruciferae* families. The results showed that TMV is distributed in all the surveyed provinces and is one of the most dominant viruses among 63 virus species detected in these four families [4]. TMV leads to one hundred million dollars losses in crops around the world in a year [5]. TMV is dependent on the plant cell to replicate and infect, which causes extreme difficulty for antiviral therapies to inhibit only the virus without damaging the host [6,7]. Therefore, the chemical method was not effective in controlling plant viruses in crop fields.

The use of pesticides has brought with it a host of issues, like the increase in drug resistance of plant pathogens, environmental pollution, and health risks to animals and humans [8,9]. In recent years, more and more people have begun to focus on the use of botanical pesticides, which display great development potential for controlling plant viral diseases, because they have low reside and are environmentally safe, biodegradable, and safe to non-target organisms [10,11]. Up until now, many kinds of plant compounds have already been demonstrated to have anti-viral ability, such as *Amaranthaceae*, *Nyctaginaceae*, *Asteraceae*, *Chenopodiaceae*, *Asclepiadaceae*, *Polygonaceae*, *Simaroubaceae*, *Acanthaceae*, *Liliaceae*, *Cruciferae*, *Leguminosae sp.*, *Boraginaceae*, *Oleaceae*, *Taxaceae*, *Ranunculaceae*, *Juglandaceae*, *Saxifragaceae*, *Theaceae*, *Schisandraceae*, *Cupressaceae*, *Labiatae*, and *Caryophyllaceae* [12,13,14,15].

Among these plants, the effective antiviral compounds are mainly proteins, alkaloids, flavonoids, phenols, essential oils, and polysaccharides. In China, four reported plant-derived ingredients have been widely used in viral disease control, including oligosaccharides, rhyscion, matrine, and fatty acids [10]. Many studies have reported the inhibitory effects of plant-derived antiviral pesticides on TMV. Tagitinin C (Ses-2) and 1β-methoxydiversifolin-3-0-methyl ether (Ses-5), two sesquiterpenoids isolated from *Tithonia diversiflia*, were found to have higher inhibitory activities than the control agent ningnanmycin [16]. Wang et al. found that sulfated lentinan induced systemic and long-term protection against TMV in tobacco [17].

*Cnidium monnieri* (L.) Cusson is a traditional Chinese medicine that is widely distributed throughout China. Many studies have suggested that it has pharmacological functions, such as anti-allergic, antipruritic, antibacterial, antidermatophytic, anti-osteoporotic, and antifungal activities [18,19,20,21,22]. *C. monnieri* was reported to contain a number of biologically active compounds such as osthole, imperatorin, bergapten, isopimpinellin, xanthotoxol, xanthotoxin, cnidimonal and cnidimarin, glucosides, sesquiterpenes, etc. [23,24,25]. The anti-viral activity of ethanol extracted from *C. monnieri* in plants remains unknown. 

In this study, we performed the osthole isolation from *C. monnieri*. We investigated whether the exogenous application of osthole is able to induce anti-viral activity in the tobacco plant when infected with TMV. The inhibitory, protective, and curative effects on TMV infection were measured. Furthermore, we observed whether osthole could affect the TMV particles and coat protein (CP) accumulation.

## 2. Results

### 2.1. Compound Structure of Osthole

Osthole (7-methoxy-8-isopentenylcoumarin): is a white solid with the molecular formula C_15_H_17_O_3_, as identified by high-performance liquid chromatography (HPLC) (Figure 1), proton nuclear magnetic resonance (^1^H-NMR), carbon-13 nuclear magnetic resonance (^13^C-NMR) (Figure 2), and high-resolution mass spectrometry (HR-MS) spectra (Figure 3). As indicated by Figure 1, the purity was greater than 98%. The spectral data was identical to that previously reported in the literature [26].

1H-NMR (500 MHz, DMSO-*d*6): *δ* (ppm) 1.61 (s, 3H), 1.71 (s, 3H), 3.40 (d, *J* = 7.2 Hz, ^2^H), 3.89 (s, 3H), 5.11–5.14 (m, ^1^H), 6.26 (d, *J* = 9.6 Hz, ^1^H), 7.05 (d, *J* = 8.4 Hz, ^1^H), 7.55 (d, *J* = 8.4 Hz, ^1^H), 7.96 (d, *J* = 9.6 Hz, ^1^H); ^13^C-NMR (125 MHz, DMSO-*d*6): *δ* (ppm) 18.1, 21.9, 25.9, 56.7, 108.5, 112.7, 113.1, 116.6, 121.7, 127.6, 132.2, 145.1, 152.6, 160.2, 160.7; HR-MS (ESI): *m*/*z* calculated for C_15_H_17_O_3_ ([M + H]^+^) 245.1170, found 245.1169. For the NMR data please see the Appendix A.

### 2.2. Anti-TMV Activities of Osthole

The anti-TMV activity of osthole from *C. monnieri* (L.) Cusson was tested at a concentration of 5 mg/mL in *N. glutinosa* using the half-leaf method. Based on the inhibition rates of local lesions on the leaves of *N. glutinosa* (Figure 4), the antiviral activity of osthole was shown to be superior to that of eugenol and ningnanmycin (Table 1), with an inhibitory effect of 72.57%, protective effect of 70.26%, and curative effect of 61.97%. 

### 2.3. The Effect of Osthole on Viral Particles

In order to determine whether osthole could directly affect the viral particle, TMV particles were mixed with the osthole at 3 mg/mL and 5 mg/mL with an equal volume for 45 min at room temperature. We found that non-treated TMV particles as observed by a Hitachi H-600 Electron Microscope appeared normal and baculiform (Figure 5A). In contrast, those treated with osthole presented with a strong detrimental effect on the virus particles (Figure 5B,C). The virus particles were gradually destroyed: as the concentration of osthole increased, the more severely the virus particles were damaged.

### 2.4. Kinetic Analysis of the Effect of Osthole Against TMV Infection

To examine which concentration of osthole could be most effective against TMV infection, a kinetic analysis was performed. The level of CP was detected by Western blot. We found that the CP level was significantly reduced when the concentration of osthole increased from 1 to 7 mg/mL in treated plants, as compared to that of the control (Figure 6). Osthole at 7 mg/mL could completely inhibit CP accumulation in TMV. This result indicates that osthole may inhibit the replication of TMV in plants. As shown in Table 2, we found that there was a significant positive correlation between the concentration of osthole and inhibitory effects on TMV infection.

## 3. Discussion

In this study, osthole was isolated from *C. monnieri* with 98% purity. Osthole is a coumarin compound, a kind of secondary metabolite in plants, which has been shown to play an important role in plant defense responses [27]. Additional properties of osthole include antibacterial, antifungal, and pesticidal functions [28]. It was reported that osthole exhibits a wide range of inhibition in mycelial growth against many fungal diseases (*Rhizoctonia solani*, *Macrophoma kawatsukai*, and *Fusarium graminearum*) [28]. However, its anti-viral activity against plant viruses has not been reported. In this study, we found that osthole has stronger anti-viral activity than eugenol and ningnanmycin. 

Furthermore, we evaluated whether osthole could directly inhibit viral particles. Through observation of TMV particles using a Hitachi H-600 Electron Microscope, we found that the compound could directly affect the particles; TMV particles were gradually destroyed. When the osthole concentration increased, the more severely the viral particles were damaged. Many reports have indicated that many changes occur in the morphology of TMV particles after treatment with plant extracts. Wang et al. found that TMV particles treated with eugenol showed ruptures and abnormality [29]. Particles were destroyed and shortened by treated with *Eupatorium adenophorum* leaf extract as reported by Jin et al. (2014) [30]. These results suggest that the method underlying viral particle destruction is a common mechanism by which plant-derived reagents act on viral infection. 

CP is critical for systemic infection and viral replication, protecting nucleic acid from enzymatic degradation, which is related to the long-distance movement of TMV and the expression of host symptoms [31,32,33]. In this study, we found the level of CP was significantly reduced to varying degrees when the concentration of osthole increased in treated plants, as compared to that of the control. Osthole at a concentration of 7 mg/mL completely inhibited expression of the TMV CP. The present results are in agreement with those reported by Li et al. (2007), Wang et al. (2014), and Chen et al. (2018) [7,11,17]. However, it remains to be further studied whether the function of osthole is through inhibiting CP synthesis or the stereoscopic assembly of the virus. 

In conclusion, osthole was isolated and purified from *C. monnieri*, and identified by ^1^H- and ^13^C-NMR and HR-MS. Osthole showed potent inhibitory activity against TMV infection. However, the antiviral mechanism of osthole on plant viruses remains unclear. In the future, we will examine whether osthole exerts its effect on CP synthesis or the stereoscopic assembly of TMV. This is the first published report on the anti-TMV activities of osthole.

## 4. Materials and Methods

### 4.1. Chemicals and Materials

Ningnanmycin AS (8%) was obtained from Deqiang Biology Co., Ltd. (Harbin, China.). Eugenol was purchased from Mckuin biological Co. LTD (Shanghai, China)

*C. monnieri* (L.) Cusson was purchased from the Jihetang Pharmacy (Yangling, China), and was identified by Professor Xiaoqian Mu at Northwest A&F University (Yangling, China).

TMV isolates were provided by the Laboratory of Molecular Plant Pathology, Southwest University (Chongqing, China) in the form of virus infected plants of *N. benthamiana*.

The seeds of *N. glutinosa* were provided by the Laboratory of Plant Virus, Inner Mongolia Agricultural University (Hohhot, China), and cultivated in an insect-free greenhouse at 24 ± 1 °C. The experiments were conducted when the plant had grown 5–6 leaves.

### 4.2. Virus Purification

The Gooding method [34] was used for the purification of TMV-inoculated *N. benthamiana*, and the isolates were stored at −20 °C and diluted to 50 μg/mL with 0.01 M PBS (phosphate-buffered saline) before use. Absorbance values were estimated at 260 nm by using an ultraviolet spectrophotometer Tu-1901 (Beijing General Instrument co. LTD, Beijing, China). To calculate the concentration of virus, the following formula was used (Equation (1)):(1)$$\mathrm{Virus}\;\mathrm{concentration}\;=\;(A_{260}\;\times \;\mathrm{dilution}\;\mathrm{ratio})/E_{1\;\mathrm{cm}}^{0.1\%,260\;\mathrm{nm}}$$

### 4.3. Isolation and Purification of Active Compounds and Structure Analysis

The *C. monnieri* (100 g) was powdered and extracted with 500 mL 90% methanol by reflux three times (1.5 h each). The combined methanol extract was concentrated (50 g) and incubated with quicklime (100 g) for 24 h. Subsequently, it was washed three times using five times diluted hydrochloric acid and concentrated, suspended in chloroform (90 mL), and isolated with alkaline water (0.5% NaOH solution). The crystal was filtered from the solution by adjusting the pH value to 7. After drying at a low temperature, the solution that was filtered from the crystal with petroleum for 1 h was naturally cooled to room temperature, and the final product was analyzed by liquid chromatography-mass spectrometry (LC-MS). The structure was identified by ^1^H- and ^13^C-NMR spectra.

### 4.4. Inhibitory Effect of Osthole on TMV Infection

The compounds of osthole and eugenol were dissolved in DMSO (dimethyl sulfoxide) (1000 mg/mL) and diluted to the required concentration with Tween-20 and distilled water (1:1000 *v*/*v*). Ningnanmycin (8%) was diluted with water to a concentration of 500 μg/mL and used for the following experiment. The inhibitory, protective, and curative effects were examined using the half-leaf method.

Osthole, eugenol, and ningnanmycin were mixed with the virus (TMV at 6 × 10^−3^ mg/mL) at the same volume or concentration for 10 min, and then were smeared with a cotton swab onto the left leaves of tobacco (*N. glutinosa*) along the main vein, whereas the virus sap and the DMSO solvent in the right half of the leaves were inoculated in each of the three treatment groups (total of 12 leaves). Each half of the leaf was smeared with 40 µL of TMV extract, and each inoculated leaf was washed with water after 10 min. The local lesion numbers were recorded for 3–4 days after inoculation and each compound and control agent was repeated three times.

The inhibition rates of osthole, magnolol, honokiol, and ningnanmycin were recorded and calculated according to the following formula (Equation (2)):
Inhibition rate (%) = [(C − T)/C] × 100%
(2)
where C is average lesion number of the control halves and T is the average mean lesion number on the drug-treated half-leaves.

### 4.5. Protective Effect of Osthole on TMV Infection

Osthole, eugenol, and ningnanmycin were gently smeared with cotton swabs on the left side of the leaves. The DMSO solution was spread as a negative control in the right lobe of tobacco leaves of the same ages. After 24 h, 40 µL of TMV (50 µL/mL) was inoculated onto whole leaves of *N. glutinosa*, each dealing with three treatment groups repeated in triplicate (total of 12 leaves were recorded), and each inoculated leaf was washed with water after 10 min. The number of lesions on tested leaves was investigated for 3–4 days.

### 4.6. Curative Effect of Osthole on TMV Infection

TMV (6 × 10^−3^ mg/mL) was inoculated on the whole leaves of *N. glutinosa* by cotton swabs. Then, the leaves were washed with water and dried. After 24 h, osthole, eugenol, and ningnanmycin were smeared onto the left leaf side, while the DMSO solution was smeared onto the right side for the control. The local lesion numbers were recorded for 3–4 days after viral inoculation. Each experiment was repeated three times.

### 4.7. Viral Particle Observation by Transmission Electron Microscope (TEM)

TMV particles were mixed with the osthole at 3 and 5 mg/mL with an equal volume for 60 min at room temperature [30]. Then, samples were placed on a carbon-coated grid, and negatively stained with a few drops of 2% phosphotungstic acid for 1 min at room temperature. They were then washed and the excess fluid was absorbed on filter paper. The samples were observed with an electron microscope (H-500, Hitachi Co. Ltd., Tokyo, Japan). An untreated virus sample served as a negative control.

### 4.8. Western Blot Analysis to Detect the CP of TMV

The levels of the TMV CP were analyzed by Western blotting. Total protein was extracted from leaves of *N. glutinosa* (0.1 g, fresh weight), that were treated by water, and 1, 3, 5, and 7 mg/mL of osthole. Samples were ground in liquid nitrogen and dissolved in 200 µL extraction buffer (125 mM Tris-HCl, pH 7.5, 2% SDS, 6 M UREA, 5% β-mercaptoethanol and bromophenol blue). The extracts were then heated at 95 °C for 10 min and centrifuged at 12,000× *g* at 4 °C for 10 min. Equal sample volumes (5 μL) were loaded on a 12% polyacrylamide gel, and proteins were separated by electrophoresis at 120 V for 70 min. After being transferred to a PVDF membrane, CP was detected using a primary antibody (1:800) and was subsequently probed with AP-coupled goat anti-rabbit IgG (1:5000; Abcam, Cambridge, UK). The signals on the membrane were visualized using Clarity Western ECL Substrate (Bio-Rad Company, Hercules, CA, USA).

### 4.9. Statistical Analysis

All data were expressed as the mean ± SD by measuring three independent replicates. The Data Processing System 15.10 (Hefei, China) was used to perform the statistical analysis. The significance of the statistical differences between three means was determined using Duncan’s new complex range method at the 5% level.

## Figures and Tables

**Figure 1 molecules-25-00065-f001:**
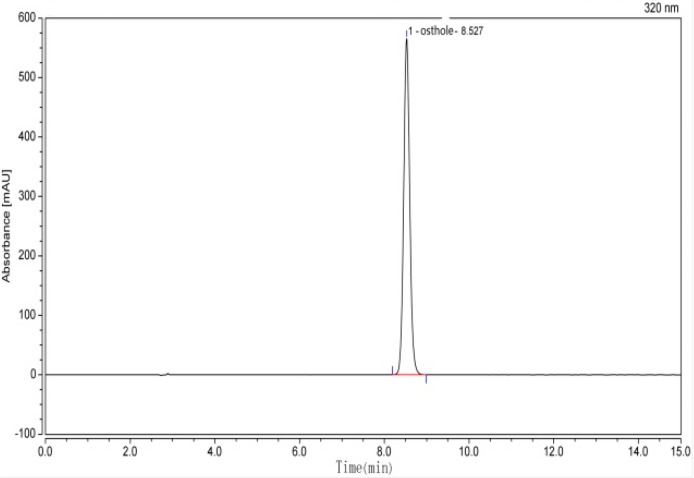
The HPLC chromatogram of osthole.

**Figure 2 molecules-25-00065-f002:**
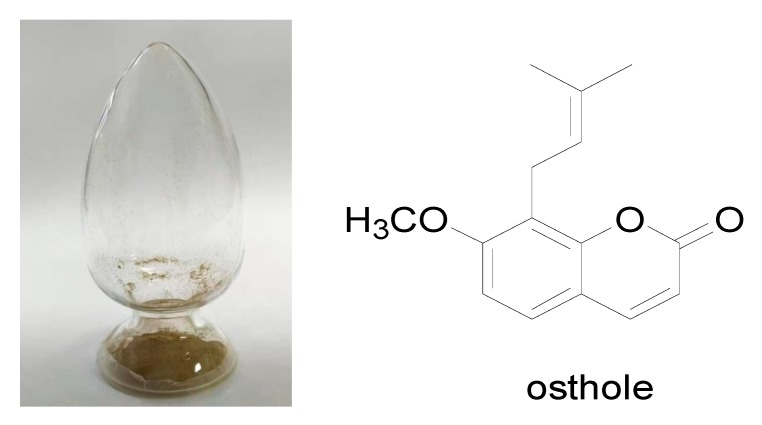
Samples and structure of the compound identified from osthole.

**Figure 3 molecules-25-00065-f003:**
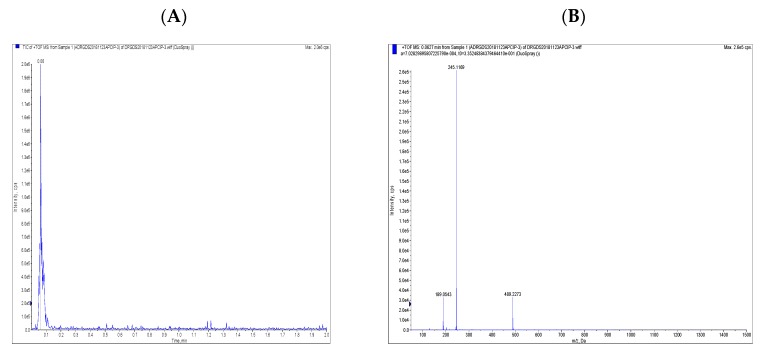
The high-performance liquid chromatography/mass spectrometry (HPLC/MS) chromatogram of osthole. (**A**) The HPLC/MS of chromatogram of osthole. (**B**) The MS of chromatogram of osthole.

**Figure 4 molecules-25-00065-f004:**
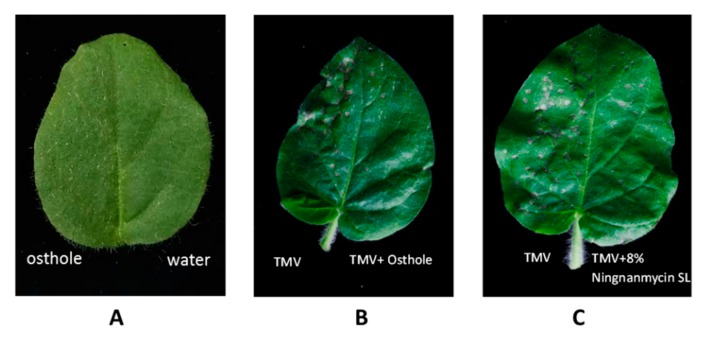
Anti-tobacco mosaic virus (TMV) activities of osthole in *N. glutinosa*. The half-leaf was smeared with osthole extract mixed with TMV at the same volume, and the right half-leaf was smeared with 40 µL of TMV. (**A**) Osthole extract (5 mg/mL) and water. (**B**) Osthole extract (5 mg/mL) and TMV. (**C**) Ningnanmycin SL (8%; 1000-X dilution) and TMV.

**Figure 5 molecules-25-00065-f005:**
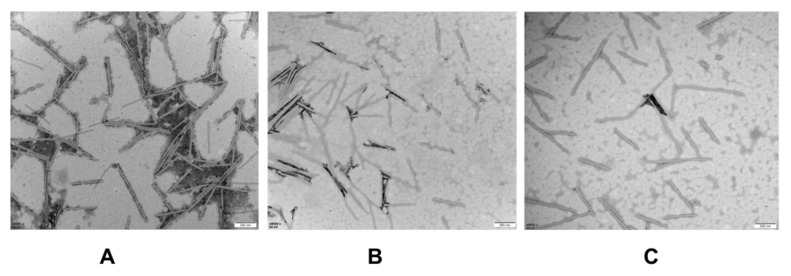
Electron microscopic observation of TMV particles after treatment with osthole for 45 min. The concentration of the purified TMV was 0.60 mg/mL. The sample was observed under 49,000× magnifications using a Hitachi H-600 Electron Microscope. (**A**) Normal TMV particles. (**B**) TMV treated with osthole at 3 mg/mL for 45 min. (**C**) TMV treated with osthole at 5 mg/mL for 45 min.

**Figure 6 molecules-25-00065-f006:**
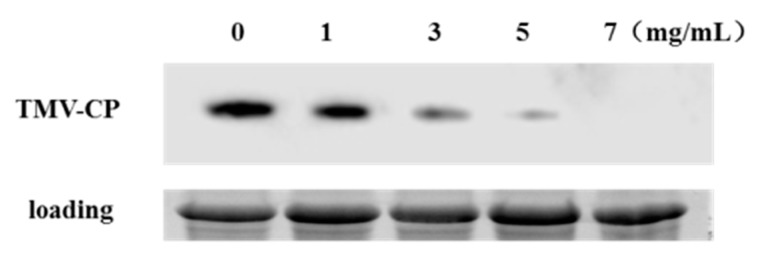
TMV coat protein (CP) accumulation detected by Western blot analysis.

**Table 1 molecules-25-00065-t001:** The anti-viral activity of osthole against TMV.

Drug	Inhibitory Effect (%)	Protective Effect (%)	Curative Effect (%)
Osthole	72.57 ± 9.24 ^aA^	70.26 ± 10.49 ^aA^	61.97 ± 7.84 ^aA^
Eugenol	60.39 ± 5.48 ^aA^	56.04 ± 4.98 ^aA^	60.83 ± 4.49 ^bB^
8% Ningnanmycin SL (1000-X dilution)	64.11 ± 2.43 ^aA^	60.57 ± 7.24 ^aA^	55.45 ± 10.96 ^aA^

Values are presented as the mean ± SE. Different upper and lower letters in the same column indicate significant difference at *p* < 0.01 or *p* < 0.05 level by Duncan’s new multiple range test.

**Table 2 molecules-25-00065-t002:** The anti-viral activity of osthole against TMV at varying concentrations.

Concentration (mg/mL)	Inhibitory Effect (%)
1	34.46 ± 5.19 ^cC^
3	53.23 ± 3.13 ^bB^
5	72.57 ± 9.24 ^aA^
7	83.22 ± 3.68 ^aA^

Data in the table are mean ± SD. Different upper and lower letters in the same column indicate significant difference at *p* < 0.01 or *p* < 0.05 level by Duncan’s new multiple range test.

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
