# Peer review of "Inhibitory Effect of Osthole from Cnidium monnieri on Tobacco Mosaic Virus (TMV) Infection in Nicotiana glutinosa"

_molecules, 2019, doi:10.3390/molecules25010065_

Round 1

Reviewer 1 Report

This work is based on the highlighting of the inhibitory effect of Osthole on Tobacco mosaic virus. The different parts of the paper are well described and documented. However, I recommand some small changes:

lines 92-94: put the latin name in italic lines 96-97: please rephrase the sentence because one can imagine that the inhibitory effects on TMV infection are concentration-dependant whereas this is not what the table shows

Author Response

Ming-Min Zhao

College of Horticulture and Plant Protection, Inner Mongolia Agricultural University,

Hohhot, Inner Mongolia, China

Tel: +86-158-2909-7529; Fax: +86-158-2909-7529

E-mail address: [email protected]

                                                                                                         Dec. 17, 2019

Dear Editor,

     We would like to resubmit the revised manuscript entitled “Inhibitory Effect of Osthole from Cnidium monnieri on Tobacco mosaic virus (TMV) Infection in Nicotiana glutinosa” to your journal, “Molecules”. We sincerely thank for the reviewers comments, which make a great improvement for our manuscript. We were also very pleased to see that all reviewers recognized the novelty and potential significance of our work. We have added some new data supposed by reviewers, described in detail below, and revised the manuscript to address reviewers’ comments. Here are our point-by-point responses:

Response to Reviewer 1 Comments

Point 1:

lines 92-94: put the latin name in italic.

Response:

Considering the Reviewer’s suggestion, the manuscript has been corrected.

Point 2:

lines 96-97: please rephrase the sentence because one can imagine that the inhibitory effects on TMV infection are concentration-dependant whereas this is not what the table shows   

Response:

We have revised this part according to the Reviewer’s suggestion.

The anti-TMV activity of osthole from C. monnieri (L.) Cusson was tested at a concentration of 5 mg/mL in N. glutinosa using the half-leaf method. Based on the inhibition rates of local lesions on the leaves of N. glutinosa (Figure. 4), antiviral activity of osthole was shown to be superior to that of eugenol and ningnanmycin (Table 1), with an inhibitory effect of 72.57%, protective effect of 70.26% and curative effect of 61.97%.

Thank you for your consideration of our manuscript.

Yours sincerely,

Yahan Chen, Ph.D.

Reviewer 2 Report

In this manuscript authors investigated the antiviral effect of osthole. Osthole was purified from C. monnieri, then the exact molecular formula was determined. Authors found a strong antiviral activity against TMV of osthole, which was more pronounced than that of eugenol and ningnanmycin. Kinetic analysis showed a concentration dependent antiviral activity. Interestingly authors found an inverse correlation between osthole concentration applied and the amount o coat protein in the TMV infected leaves. And also increasing concentration of osthole caused the reduction of intact TMV particles determined by electron microscopy.

In summary in my opinion authors presented solid results proving the antiviral effect of osthole.

I only have one demand. In Fig. 4. The effect of osthole could be clearly seen in TMV infected leaves. I am asking the authors to provide a panel showing the effect of osthole to uninfected leaves. That would serve as an important control.

Author Response

Ming-Min Zhao

College of Horticulture and Plant Protection, Inner Mongolia Agricultural University,

Hohhot, Inner Mongolia, China

Tel: +86-158-2909-7529; Fax: +86-158-2909-7529

E-mail address: [email protected]

                                                                                                         Dec. 17, 2019

Dear Editor,

     We would like to resubmit the revised manuscript entitled “Inhibitory Effect of Osthole from Cnidium monnieri on Tobacco mosaic virus (TMV) Infection in Nicotiana glutinosa” to your journal, “Molecules”. We sincerely thank for the reviewers comments, which make a great improvement for our manuscript. We were also very pleased to see that all reviewers recognized the novelty and potential significance of our work. We have added some new data supposed by reviewers, described in detail below, and revised the manuscript to address reviewers’ comments. Here are our point-by-point responses:

Response to Reviewer 2 Points

Point 1:

I only have one demand. In Fig. 4. The effect of osthole could be clearly seen in TMV infected leaves. I am asking the authors to provide a panel showing the effect of osthole to uninfected leaves. That would serve as an important control.

Response

Considering the Reviewer’s suggestion, the manuscript has been corrected.

Figure 4. Anti-TMV activities of osthole in N. glutinosa. The half leaf was smeared with osthole extract mixed with TMV at the same volume, and the right half leaf was smeared with 40 µL of TMV. (A) Osthole extract (5 mg/mL) and water; (B) Osthole extract (5 mg/mL) and TMV; (C) Ningnanmycin SL(8%; 1,000-X dilution) and TMV.

Sincerely thanks for your nice Points.

Yours sincerely,

Yahan Chen, Ph.D.
